# Impact of Oxalate Ligand in Co-Precipitation Route on Morphological Properties and Phase Constitution of Undoped and Rh-Doped BaTiO_3_ Nanoparticles

**DOI:** 10.3390/nano9121697

**Published:** 2019-11-28

**Authors:** Roussin Lontio Fomekong, Shujie You, Francesco Enrichi, Alberto Vomiero, Bilge Saruhan

**Affiliations:** 1Department of High-Temperature and Functional Coatings, Institute of Materials Research, German Aerospace Center, 51147 Cologne, Germany; 2Division of Materials Science, Department of Engineering Sciences and Mathematics, Luleå University of technology, 97187 Luleå, Sweden; shujie.you@ltu.se (S.Y.); francesco.enrichi@unive.it (F.E.); alberto.vomiero@ltu.se (A.V.); 3Department of Molecular Sciences and Nanosystems, Ca’ Foscari University of Venice, Via Torino 155, 30172 Venezia Mestre, Italy

**Keywords:** Rh-doped BaTiO_3_, nanoparticles, synthesis route, co-precipitation, perovskite

## Abstract

In order to design and tailor materials for a specific application like gas sensors, the synthesis route is of great importance. Undoped and rhodium-doped barium titanate powders were successfully synthesized by two routes; oxalate route and classic route (a modified conventional route where solid-state reactions and thermal evaporation induced precipitation takes place). Both powders were calcined at different temperatures. X-ray diffraction (XRD), Raman spectroscopy, scanning electron microscopy (SEM), energy-dispersive x-ray spectroscopy (EDX) and Brunauer-Emmet-Teller (BET) analyses are employed to identify the phases and polymorphs, to determine the morphology, the chemical composition and the specific surface area of the synthesized materials, respectively. The so-called oxalate route yields pure BaTiO_3_ phase for undoped samples at 700 °C and 900 °C (containing both cubic and tetragonal structures), while the classic route-synthesized powder contains additional phases such as BaCO_3_, TiO_2_ and BaTi_2_O_5_. Samples of both synthesis routes prepared by the addition of Rh contain no metallic or oxide phase of rhodium. Instead, it was observed that Ti was substituted by Rh at temperatures 700 °C and 900 °C and there was some change in the composition of BaTiO_3_ polymorph (increase of tetragonal structure). Heat-treatments above these temperatures show that rhodium saturates out of the perovskite lattice at 1000 °C, yielding other secondary phases such as Ba_3_RhTi_2_O_9_ behind. Well-defined and less agglomerated spherical nanoparticles are obtained by the oxalic route, while the classic route yields particles with an undefined morphology forming very large block-like agglomerates. The surface area of the synthesized materials is higher with the oxalate route than with the classic route (4 times at 900 °C). The presence of the oxalate ligand with its steric hindrance that promotes the uniform distribution and the homogeneity of reactants could be responsible for the great difference observed between the powders prepared by two preparation routes.

## 1. Introduction

Barium titanate (BaTiO_3_) has a typical perovskite structure which exists in a number of polymorphs from high to low temperature: hexagonal, cubic, tetragonal, orthorhombic, and rhombohedral crystal structure. All polymorphs are related to the cubic ABO_3_ perovskite structure which consists of close-packed AO_3_ layers stacked along [001] in a cubic sequence (i.e., ABCABC) with B cations occupying octahedral sites to form a lattice of corner-linked BO_6_ octahedra [1]. The tetragonal polymorph of BaTiO_3_, in which the Ti^4+^ cations are displaced from the centre of the octahedra towards an apex, has received much attention as a result of the high permittivity at the tetragonal–cubic phase transition [2]. Undoped BaTiO_3_ is stable in the hexagonal polymorph at high temperatures (above 1460 °C) and adopts this structure until it melts at 1620 °C [3]. This hexagonal structure can be stabilized at lower temperatures by a number of B site dopant cations, of comparable size to Ti^4+^.

Relying on excellent dielectric, ferroelectric, piezoelectric and optical properties, BaTiO_3_ is a well-known electro-ceramic material that is used in the manufacturing of thermistors, dielectric ceramic capacitors [4,5,6] and also applied as photocatalyst and gas sensor material [7,8]. These properties of BaTiO_3_ are strongly influenced by the existing polymorphs which are linked to the preparation method as well as the dopant. The gas sensor application also depends on the particle size and morphology. In fact, for sensor application, the material needs to adsorb a large quantity of gas on its surface requiring a high surface area and channel-like pore structure. The surface area depends on the particle size and the morphology. The fine spherical morphology offers sufficient surface for gas adsorption, but, must not sinter and cause pore closure. Moreover, the dopant plays a key role in gas sensing (as well as in catalytic NO_x_-reduction). Rh dopant contributes in superficial catalytic reactions and can considerably decrease the baseline resistance due to its metallic nature. BaTiO_3_ being a dielectric results in a high baseline resistance leading to measurement difficulties in gas sensing. The dopant can also act as a catalyst and, thus, enhance the selectivity of sensing material for a specific gas. During the last decade many efforts have been devoted to the preparation of metal-incorporated or doped BaTiO_3_ using different synthesis techniques. These can be divided in two categories: (i) solid-state reaction methods and (ii) wet chemistry methods. The first consists of mixing two solid precursors (e.g., TiO_2_ and BaO, or TiO_2_ and BaCO_3_) and heat treatment of the mixture at high temperatures (≥1000 °C). For instance Ni-doped BaTiO_3_ and Mn-doped BaTiO_3_ have been prepared by a solid-state reaction method at 1300 °C [9,10]. The second method includes hydrothermal, sol-gel, co-precipitation and other techniques. The main advantage of the wet chemistry synthesis techniques is the quasi-atomic dispersion of the dopant as well as Ba and Ti in liquid precursors, which facilitates synthesis of the crystallized powder with nanometer particle sizes usually at low temperatures (<800 °C). Fe-doped BaTiO_3_ has been prepared by the hydrothermal route [11], Co-doped BaTiO_3_ by sol-gel route [12] and Rh-doped BaTiO_3_ by hydrothermal and polymerized complex (PC) methods [13,14]. The synthesis route plays a very important role on the final properties of the materials [15,16,17] since the morphology, particle size, polymorphs and phase compositions are related directly to the synthesis method and also reflect the desired functional property of the material.

Therefore, this work investigates, to the best of our knowledge for the first time, the effect of the oxalate ligand in co-precipitation route on morphological properties and phase constitution of undoped and Rh-doped BaTiO_3_. For that, undoped and Rh-doped BaTiO_3_ is synthesized by co-precipitation method using oxalate as the precipitating agent (achieved for the first time and named hereafter as ‘oxalic route’) and compared with that synthetized by applying a modified solid-state route (hereafter named as ‘classic route’). The effect of these two routes as well as the decomposition temperature on the phase formation, polymorph, morphology, texture and particle size of the as synthesized materials are investigated and the results are compared.

## 2. Materials and Methods

### 2.1. Materials for Synthesis of Rh-Doped BaTiO_3_

Barium acetate (Ba(C_2_H_3_O_2_)_2,_ 98% Chempur, Karlsruhe, Germany), titanium isopropoxide (Ti{OCH(CH_3_)_2_}_4_, 99.999%, Aldrich, Steinheim, Germany), and rhodium nitrate (Rh(H_2_O)(OH)_3-y_(NO_3_)_y_ y = 2–3, 15.19%, Chempur, Karlsruhe, Germany), oxalic acid (C_2_H_2_O_4_ 2H_2_O, 99.5%, Chempur, Karlsruhe, Germany), acetic acid (99.9%, Chemsolute, Renningen, Germany) and absolute ethanol (Chemsolute, Renningen, Germany) were used in the forms as received from suppliers without further purification or processing.

#### 2.1.1. Synthesis by the Modified Solid State Reaction Methods (Classic Route)

The undoped barium titanate (named BT-CL-700/900/1000) and the rhodium-doped barium titanate (BTR1-CL-700/900/1000) powders were synthesized by the same procedure (Figure 1a). For the preparation of the Rh-doped sample, firstly barium acetate was dissolved in acetic acid as titanium isopropoxide was diluted with ethanol solution to obtain barium and titania sols. These sols were mixed to yield a homogenous mixed-sol into which the Rh-nitrate solution that is obtained by diluting in ethanol solution was added. The resulting yellowish solution was then stirred at 80 °C till the total evaporation of the solvent occurred. The amount of the aqueous Rh-nitrate solution was adjusted to yield perovskites with the following composition: BaTi_0.98_Rh_0.02_O_3_. The solid residue obtained after the evaporation of the solvent was dried at around 100 °C on a hot plate and then finally calcined in a furnace under static air at 700 °C, 900 °C and at 1000 °C for 1 h with a heating rate of 5 °C min^−1^. 

#### 2.1.2. Synthesis by the Oxalate Coprecipitation Route

The same processing steps are employed for synthesis of undoped barium titanate (hereafter named BT-OX-700/900/1000) and rhodium-doped barium titanate (hereafter named BTR1-OX-700/900/1000). The synthesis of Rh-doped BaTiO_3_ resumed in Figure 1b was followed over two steps: the first step involved the preparation of a precursor solution containing barium, titanium and rhodium ions in the right ratio. This was then co-precipitated by using an oxalate ligand as precipitating agent which was subsequently decomposed by heating. Barium acetate was dissolved in acetic acid, titanium iso-propoxide and rhodium nitrate were diluted separately by adding ethanol. All the prepared solutions were mixed and stirred for 5 min to obtain a yellowish solution. The oxalic acid was dissolved in ethanol solution and poured progressively in the previously prepared cationic solution. The resulting mixture was stirred for 1 h to allow the complete precipitation. The precipitate that was obtained was filtered and dried in the oven at 80 °C yielding a yellowish powder. The as-prepared precursor powder was calcined in ceramic combustion boat holder at 700 °C, 900 °C and 1000 °C in a muffle furnace (5 °C min^−1^) for 1 h in static air. 

### 2.2. Characterization

The X-ray diffraction (XRD) diffractograms of all the samples were collected at room temperature with a D5000 Siemens Kristalloflex θ–2θ powder diffractometer (Siemens, Munich Germany) which has a Bragg-Brentano geometry using Cu-Kα radiation (λ = 1.54178 Å) and a standard scintillation counter detector. For the experiments, the as prepared material was pressed in sample holder and the patterns were recorded in the range of 5–80° with a scan step of 0.02 (2θ) and a 2 s step^−1^. The reflections from the JCPDS (Joint Committee on Powder Diffraction Standards) database were assigned to the experimental diffractograms with the program EVA by BRUKER AXS. The recorded patterns were also indexed and the unit cell refined by using the STOE WinXPOW software package (STOE WinXPOW Version 1.10; STOE&Cie GmbH, Darmstadt, Germany, 2002).

All the Raman spectra presented here were recorded at room temperature on Bruker Senterra Raman spectrometer (Bruker Optik GmbH, Ettlingen, Germany) under 532 nm and 0.2 mW power laser excitation which was focused on samples through a 50× objective (Olympus MPlan N 50×/0.75).

The morphology of the particles was determined by scanning electron microscopy (SEM) analysis and was carried out in a Zeiss Ultra 55 microscope (Zeiss, Jena, Germany) equipped with an energy-dispersive X-ray spectrometer (EDX) from Oxford Instruments (Oxford, England). Prior to the analysis, the samples were sputtered with thin particulate Platine layer to avoid charge effects. The semi-quantitative elemental analysis of the sample was performed by the energy-dispersive X-ray (EDX) method. For that, the specimens were analyzed by performing the experiment at 15 keV with a working distance of 8 mm. The acquisition time for the chemical spectra lasted 300 s with a probe current of 1 nA. The quantitative analysis of the atomic elements was achieved with the integrated Aztec software (AZtec 4.1 SP1, Oxford, England).

Specific surface was determined using Micromeritics Tristar 3000 equipment (Micromeritics, Aachen, Germany) using N_2_ adsorption at 77.150 K. Samples were degassed under vacuum at 120 °C for a period of 2 h prior to surface area measurement. The specific surface area was calculated using the Brunauer-Emmet-Teller (BET) equation.

## 3. Results and Discussions

The phase compositions of all the prepared samples were investigated by XRD and are presented in Figure 2 and Figure 3 and Table 1.

As shown in Figure 1b the undoped powders synthesized via oxalate route display the presence of only the single phase, BaTiO_3_ (according JCPDS 075-0462) (Figure 2a) at 700 and 900 °C. At 1000°C, in addition to BaTiO_3_, as minor phase, orthorhombic BaTi_2_O_5_ appears. However in the samples prepared by the classic route (Figure 2b), a mixture of BaCO_3_, BaTiO_3_, orthorhombic BaTi_2_O_5_, and TiO_2_ rutile is obtained at 700 °C while at 900 °C and 1000 °C BaTiO_3_ as major phase and orthorhombic BaTi_2_O_5_ as minor phase are observed. Additionally, it is noted that single and pure phase of BaTiO_3_ can be obtained only with the oxalic route and this forms already at 700 °C. This means that the oxalic route reduces the formation temperature of the pure crystalline phase, barium titanate considerably. This may be due to the presence of oxalate ligand which allows a better distribution and homogeneity of reactant. However the formation of BaTi_2_O_5_ at 1000 °C indicates that the pure BaTiO_3_ obtained through the oxalic route is metastable and decomposes or starts to convert to BaTi_2_O_5_ phase above 900 °C. We can also observe that as the temperature increases, the XRD peak intensity increases also due certainly to the increase of crystallite size (inset in Figure 2a).

For the Rh-doped powder prepared via the oxalic route (Figure 3a), XRD results yielded the presence of the following phases after calcining at 700 °C, 900 °C: BaTiO_3_ as the major and BaCO_3_ in trace amount. However, at 1000 °C, the powder contained Ba_3_RhTi_2_O_9_ phase in addition to the previous phases obtained at lower temperature. In the presence of Rh doping, we observed the formation of BaCO_3_. The same observation has been reported in literature for Cr doped BaTiO_3_ [18]. This can be due to the fact that in those synthesis conditions, a small part of Ba is replaced by a part of Rh (which was not considered during the calculation of the amount of required reactant) and the free barium resulting from this substitution formed easily barium carbonate. This possibility of substitution was also observed in Co-doped BaTiO_3_ reported by L. Padilla-Campos et al. [19].

As the temperature increases, the amount of BaCO_3_ decreases due to the decomposition of BaCO_3_ and the decrease of the amount of Ba substituted by Rh. No evidence of an Rh phase was encountered at 700 °C and 900 °C. As shown in Figure 4a, the lattice parameter a increases, implying the cell volume is enlarged due to the larger radius of Rh^3+^ (0.67 Å) compared with Ti^4+^ (0.60 Å). This increase is higher at 700 °C than at 900 °C (from 4.011 Å to 4.028 Å at 700 °C and from 4.010 Å to 4.015 Å at 900 °C) indicating that the substitution of Ti by Rh is more pronounced at 700 °C. At 1000 °C there was almost no increase of lattice parameter (from 4.009 to 4.010 Å) (Figure 4a) and there existed another phase which contained rhodium (Ba_3_Ti_2_RhO_9_). It can be suggested that at this temperature, there is no longer substitution of titanium by rhodium. In other words, the Bragg diffraction peaks of powders calcined at this temperature are nearly coincident with the undoped samples. It can be proposed that BaTiO_3_ phase incorporating Rh^3+^ -ions display a phase change, leading to the formation of a new phase, Ba_3_Ti_2_RhO_9_, due to the diffusion of Rh^3+^-ions out of the BaTiO_3_ lattice. This might be because of the better phase stability of Ba_3_Ti_2_RhO_9_ even at a temperature of 1000 °C. 

Rh-doped powder prepared by the modified classic route (Figure 3b) contained at 700 °C: BaTiO_3_, BaCO_3_ and BaTi_2_O_5;_ at 900 °C: BaTiO_3_ and BaTi_2_O_5_ and at 1000 °C: BaTiO_3_, BaTi_2_O_5_ and Ba_3_RhTi_2_O_9_ (see Table 1). Although at 1000 °C the composition of the doped sample is similar to that obtained by oxalic route, at 700 °C and 900 °C following the classic route leads again to less pure powders due to the reason that was given previously. As shown in Figure 4b, no evidence for the increase of lattice parameter is observed at 700 °C (from 4.012 Å to 4.013 Å) and there is no decrease of the peak intensity like in oxalic route which means that there was probably no substitution of Ti^4+^ by Rh^3+^. However at 900 °C, the increase of the lattice parameter is observed (from 4.009 Å to 4.013 Å) with the decrease of peak intensity and at 1000 °C almost no change of lattice parameter is observed (from 4.008 Å to 4.009 Å). The occurrence of the Ti^4+^ substitution at lower temperatures by the oxalic route could be explained by the excellent dispersion and homogenization of Rh in the presence of oxalate ligand which in turn allows an easy substitution process. That is not the case with the classic route; therefore the substitution process requires more thermal activation.

In order to confirm the structure of the synthesized materials and to determine the type of BaTiO_3_ polymorph obtained, Raman spectroscopy measurements were carried out. Raman scattering spectra are more sensitive to short range ordering structure [20]. Therefore, while XRD measurement can clarify the average and static symmetries, Raman scattering can clarify the local and dynamic symmetries. Figure 5 shows the Raman spectra of all the samples. As shown in Figure 5a–c the Raman spectra of undoped samples prepared by oxalic route at 700, 900 and 1000 °C are different from what obtained by classic route. The samples prepared by co-precipitation yield almost identical peaks to BaTiO_3_, with the ones observed at 270, 308, 525 (weak) and 725 (weak) cm^−1^ which are attributed respectively to the A1(TO2), E(TO2), A1(TO3) and A1(LO3) of barium titanate modes of the room temperature P4mm phase [21]. The weakness of the peaks at 725 cm^−1^ indicated the presence of a low amount of tetragonal phase suggesting that we have a mixture of cubic (major) and tetragonal phases. However, the samples prepared by the classic route showed additional peaks at 700 °C and 900 °C. The additional peaks are attributed to BaCO_3_ (694, 1060 cm^−1^), BaTi_2_O_5_ (880 cm^−1^) and TiO_2_ (448, 612, 827 cm^−1^). At 1000 °C the spectra of samples obtained by the two routes are almost similar indicating the same composition. All these results are in agreement with the XRD analysis. On doping (see Figure 5d–f), the peak at 725 cm^−1^ increases, indicating the increase of tetragonal phase for the samples prepared by the oxalic route. Since Raman scattering can be observed at around 270 and 525 cm^−1^ for both tetragonal and cubic symmetries of BaTiO_3_ [22], the tetragonal phase can be determined by observing the relative intensity of the bands at 725 cm^−1^. In fact, the band at 725 cm^−1^ is related to the highest-wavenumber longitudinal optical mode (LO) of A1 symmetry. Therefore, these bands are absent in the cubic para-electric phase [23,24]. Although the observation of this band confirms the presence of the tetragonal phase, this does not exclude a possible cubic phase existence. The relatively intense band at 640 cm^−1^ after doping could be assigned to the existence of a hexagonal phase at low temperature usually justified by two Raman bands at 153 and 640 cm^−1^ [22]. In fact at these wavenumbers, the appearance of bands belonging to other barium titanate phases is not expected [25,26,27]. The band at 153 cm^−1^ is less intense and, therefore, rarely observed. The hexagonal phase of barium titanate is normally stable at high temperature (1432–1625 °C) and has a TiO_6_ octahedra configuration that is significantly different from that of the tetragonal phase. In the tetragonal phase, two TiO_6_ octahedra share only one point, while in the hexagonal phase, they share the entire surface [27]. Yashima et al. [27] explained the presence of the hexagonal phase at low temperature as the result of a reconstructive phase transition. For our case, the appearance of the hexagonal phase at low temperature can be explained by a local reducing atmosphere during the decomposition of the organic components (oxalate ligand) of the precursor [28]. In this atmosphere, the reduction of Ti^4+^ to Ti^3+^ promotes the formation of planar defects so that the stacking sequence in the regions of the defects is equivalent to that of the high-temperature hexagonal barium titanate [26,29]. In summary, the presence of dopant seems to promote the formation of tetragonal phase and induce the formation of hexagonal phase. By classic route, the peaks for doped and undoped are almost similar at 700 °C. At 900 °C, some peaks (612, 827 cm^−1^) disappear which means that the phase became purer. At 1000 °C some peaks become more intense (725, 640 cm^−1^) indicating the increase of the amount of tetragonal and hexagonal phases. We can observe that at 700 °C and 900 °C the peak at 725 cm^−1^ is already intense in undoped samples, indicating the presence of tetragonal phase which is not the case with the oxalate route.

The morphology of the particles obtained by the classic and oxalate co-precipitation routes is shown in Figure 6 and Figure 7. As it is clear at first glance, the synthesis route and the temperature do have a significant influence on the morphology. The oxalate co-precipitation route leads to homogeneously distributed nano-spherical particles for the undoped BaTiO_3_ (Figure 6b,d) and Rh-doped samples (Figure 7b,d) on calcining at 700 and 900 °C. However the classic route yields for undoped BaTiO_3_ (Figure 6a,c), large particle blocks already at 700 °C and large faceted particles with a morphology containing stacked and dense blocks formed on recrystallization at 900 °C. These particles then sintered to finer ones in the case of the Rh-doped BaTiO_3_ powders (Figure 7a,c).

The presence of a precipitating agent in the co-precipitation route is certainly responsible for the thermal decomposition driven formation of spherical and homogeneous particles. In fact, the precursor which decomposes thermally contains a mixture of barium titanium rhodium oxalate, resulting in a bonding formation between the metallic Ba^2+^ or Ti^4+^ cations and the oxalate ligand (which is not the case for the classic route). The oxalate ligand acts as buffer and prevents the stacking of the particles during thermal treatment. In other word, the presence of a precipitating agent (e.g., in this case oxalate) avoids the stacking of the particles by steric hindrance during thermal decomposition. The thermogravimetric analysis (TGA) measurement (not shown here) of the oxalate route produced powder displays that organic substances leave the system at 700 °C. In fact, the first two steps of a sintering process (particle rearrangement and particle contact through neck formation) occur before the thermal elimination of oxalate ligand is completed. Once the rigid necks are formed, there is no possibility for particles to move together. Thus, the oxalate route produced powder does not yield any volume shrinkage on further temperature increase to 1000 °C. Instead, the next stage of the sintering process, i.e., grain growth takes place. Thus, in conclusion, the oxalate processed material experiences retardation in the sintering process. Moreover, above the oxalate decomposition temperature (650–700 °C), the sintering process may proceed with the grain growth stage but owing to the space left behind from the decomposing and vaporizing organic part, the growth rate of these particles is low. This is because solid/solid interaction occurs only at the neck points yielding extension of particles through the transfer of material to the neck points. In the case of the classic route, where there was no organic ligand addition, the processes such as heavy particle rearrangement and particle pulling-together result in contact all along the particle grain boundaries. Thus, a strong shrinkage takes place followed by a solid state sintering process and grain growth. This is confirmed by different SEM pictures of the classic route where it can be observed that the particles are highly sintered together and are grown in comparison to those in the oxalate route produced material.

It is also notable that Rh-doping results in a decrease in the size of the individual particles or of the particles in a block with the oxalic route or with the classic route, respectively. This observation is in accordance with XRD where we observed the decrease of peak intensities with the doping. On further heat-treatment of the powders synthesized by the classic route at 1000 °C, the large dense blocks, built of rhombohedral crystallites are formed (Figure 6e and Figure 7e). In turn, the oxalate co-precipitation route yields only larger aggregates of spherical nanoparticles (Figure 6f and Figure 7f). At this temperature, it appears that the Rh-doping at both routes results in an enhancement of the agglomeration process. It is observed that the increase of the heat-treatment temperature leads to an increase in particle mobility resulting in particle growth and this is in agreement with XRD results.

Energy-dispersive X-ray spectroscopy (EDX) is an analytical technique used for elemental analysis. The compositions of all 4 samples (BT-OX-700, BT-CL-700, BTR1-OX-700 and BTR1-OX-700) taken as representative of all the samples are presented in Table 2. In order to check the homogeneity of the samples, the measurements were done in three different areas. As can be observed, the samples prepared by the oxalic route showed almost the same composition everywhere with Ti/Ba ratio equal roughly to 1, indicating the good homogeneity and this is not the case with the samples prepared by classic route (e.g., sample BT-CL-700). For the undoped samples prepared by the oxalic route, the mean atomic percentages obtained from EDX quantification analysis were 21.34% of Ti (cf 20% expected), 18.85% of Ba (cf 20% expected) and 59.80% for O (cf 60% expected) which corresponded to BaTiO_3_ as revealed by XRD and Raman. After doping, the percentages of Ti and Ba dropped to low values (although the ratio between Ti/Ba did not change greatly), due certainly to the presence of additional compounds which increase the total number of atom. This is confirmed by the detection of BaCO_3_ revealed by XRD after Rh doping.

The specific surface areas of the synthesized samples at 900 °C and 1000 °C taken as representative of all the samples obtained by BET measurements are presented in Table 3. These results indicate that the specific surface area of the powders synthesized by the oxalic route and calcined at 900 °C is a factor of four greater than that obtained with the powder synthesized by classic route, while this relation is reduced roughly to a factor of two after calcining the powders at 1000 °C. This difference in the surface area could be related to the morphological distribution of the particles. In fact as observed by SEM, the spherical nanoparticles obtained by the oxalic route may offer more freely available surfaces for the gas adsorption such as N_2_ during BET measurements while the stacked and dense blocks obtained with the classic route do not allow the inter-diffusion of nitrogen. It should also be mentioned that the specific surface areas of doped and undoped powders calcined at 900 °C differ, indicating an increase with the Rh-doping. This increase is in agreement with the SEM and XRD results displaying a decrease in the particle and block sizes for the Rh-doped powders synthesized with the oxalic and classic routes, respectively. The reduction of specific surface area observed with the powders calcined at 1000 °C can be explained relying on the microstructural observations by means of SEM that the particle agglomeration occurs as the heat-treatment temperature increases, leading to the reduction of surface area that is available for gas adsorption. This agglomeration is more pronounced at the powders synthesized by means of the classic route where the formation of dense blocks is obvious at the SEM micrographs. In the case of the oxalic route, the presence of precipitation agent might have limited this agglomeration, since the micrographs display a more open spaced microstructure. It should be noticed that there is a reduction of specific surface area with the doping at 1000 °C for the oxalic route-synthesized powder, due certainly to the increase of agglomeration with doping as observed on SEM.

In summary, Rh-doped BaTiO_3_ powders synthesized by the oxalic route offers the following features: (i) high phase purity, (ii) high surface area, (iii) sintering and growth resistant nano-sized spherical particles, (iv) thermal phase stability of BaTiO_3_ up to 1000 °C. These features can be effectively used to enhance the functional properties of barium titanate such as gas sensor which will be investigated in the future. 

## 4. Conclusions

The undoped and Rh-doped BaTiO_3_ have been prepared by a modified solid-state route (classic route) and the oxalic coprecipitation route and calcined at different temperatures (700 °C, 900 °C or 1000 °C). The effect of synthesis route and the calcination temperature on the phase constituents, microstructure and morphology of the as-synthesized materials have been investigated. Pure BaTiO_3_ phase, free of any other secondary phases, is obtained at 700 and 900 °C by oxalic route while secondary phases of BaCO_3_, TiO_2_ and BaTi_2_O_5_ occurs in the undoped powders synthesized by the classic route under the same conditions. The doping type of rhodium is debatable and can be assumed to be a substitution of Ti^4+^ by Rh^3+^ for both preparation routes. However, the temperature of substitution is lower by oxalic route (700 °C) than by classic route (900 °C) and the amount of substituted Ti^4+^ decreases with the increase of temperature. On calcining at 1000 °C, there occurs a phase separation leading to the displacement of rhodium into this new phase. A spherical morphology and nanoparticle sizes are observed with the oxalic route, which is not observed with the classic route. It was found that the preparation process as well as the temperature and the doping have a great influence on the phase and polymorph composition as well as the morphology and particle size of the Rh-doped BaTiO_3_. This study might be extended to other mixed metal oxide materials in order to fully understand the influence of the synthetic method on the structural, morphological, textural and functional properties of the material and by that to optimize a material for a specific application such as gas sensors.

## Figures and Tables

**Figure 1 nanomaterials-09-01697-f001:**
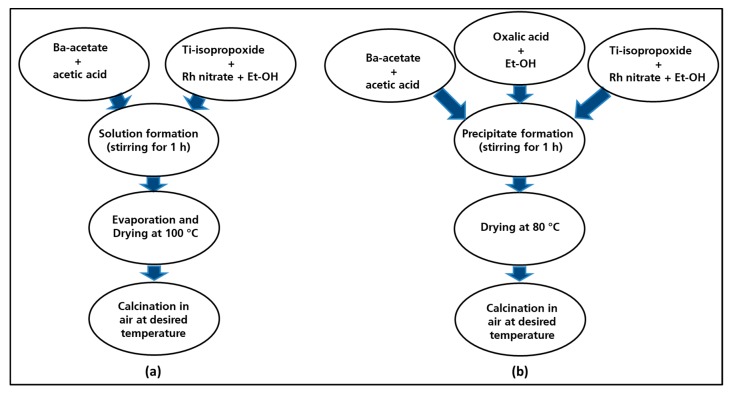
Flow diagram for the synthesis route of the materials: classic route (**a**), oxalate route (**b**).

**Figure 2 nanomaterials-09-01697-f002:**
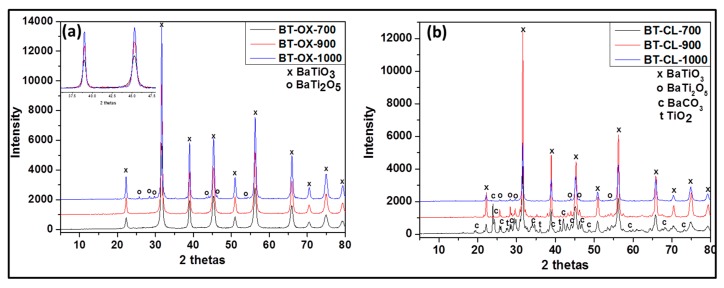
Powder X-ray diffraction (XRD) patterns of undoped barium titanate prepared via oxalate route at 700 °C (BT-OX-700), 900 °C (BT-OX-900), 1000 °C (BT-OX-1000) (**a**) and classic route at 700 °C (BT-CL-700), 900 °C (BT-CL-900) and 1000 °C (BT-CL-1000) (**b**).

**Figure 3 nanomaterials-09-01697-f003:**
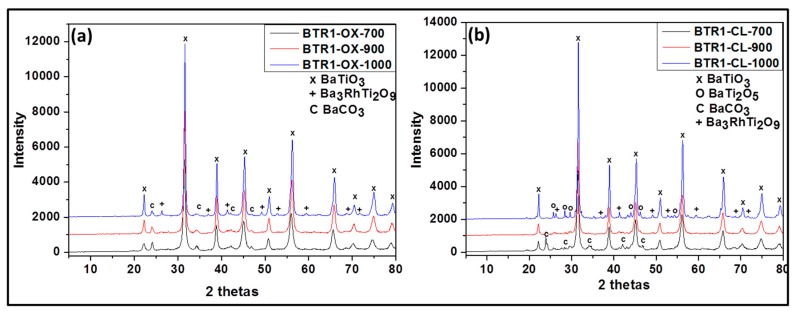
Powder XRD patterns of Rh-doped barium titanate prepared via oxalate route at 700 °C (BTR1-OX-700), 900 °C (BTR1-OX-900), 1000 °C (BTR1-OX-1000) (**a**) and classic route at 700 °C (BTR1-CL-700), 900 °C (BTR1-CL-900) and 1000 °C (BTR1-CL-1000) (**b**).

**Figure 4 nanomaterials-09-01697-f004:**
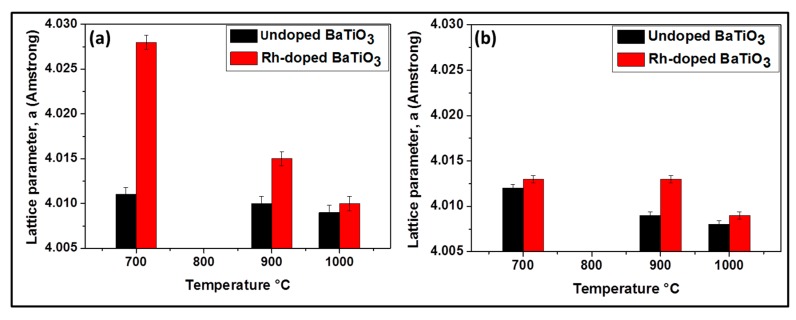
Lattice parameters of BaTiO_3_ for samples prepared by the oxalic route (**a**) and by the classic route (**b**), respectively.

**Figure 5 nanomaterials-09-01697-f005:**
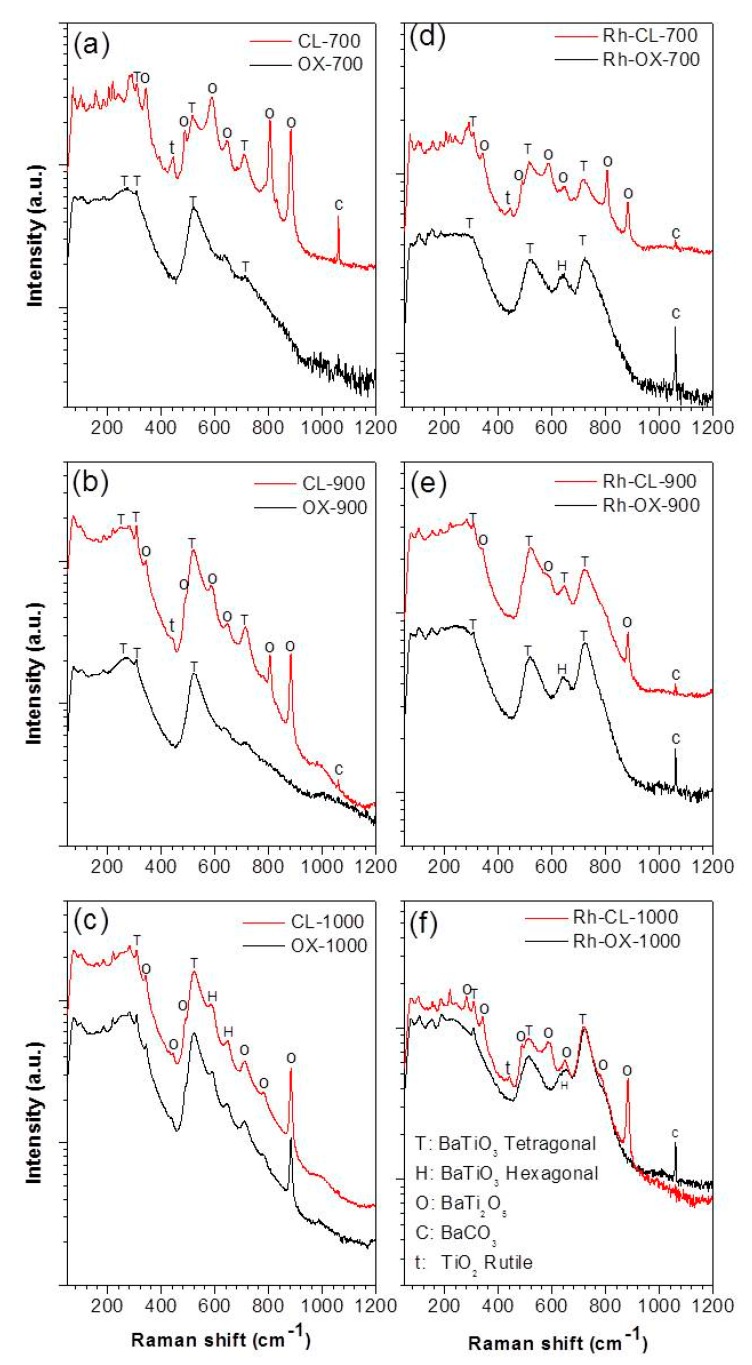
Raman spectra of undoped BaTiO_3_ (**a**–**c**) and Rh doped BaTiO_3_ (**d**–**f**) prepared by oxalic and classic routes.

**Figure 6 nanomaterials-09-01697-f006:**
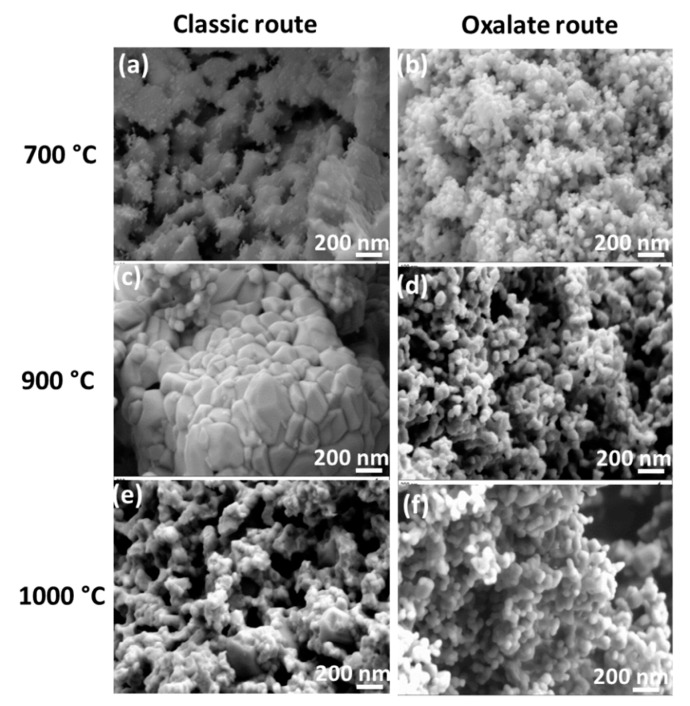
Scanning electron microscopy (SEM) images of undoped samples, BT-CL-700 (**a**), BT-OX-700 (**b**), BT-CL-900 (**c**), BT-OX-900 (**d**), BT-CL-1000 (**e**) and BT-OX-1000 (**f**).

**Figure 7 nanomaterials-09-01697-f007:**
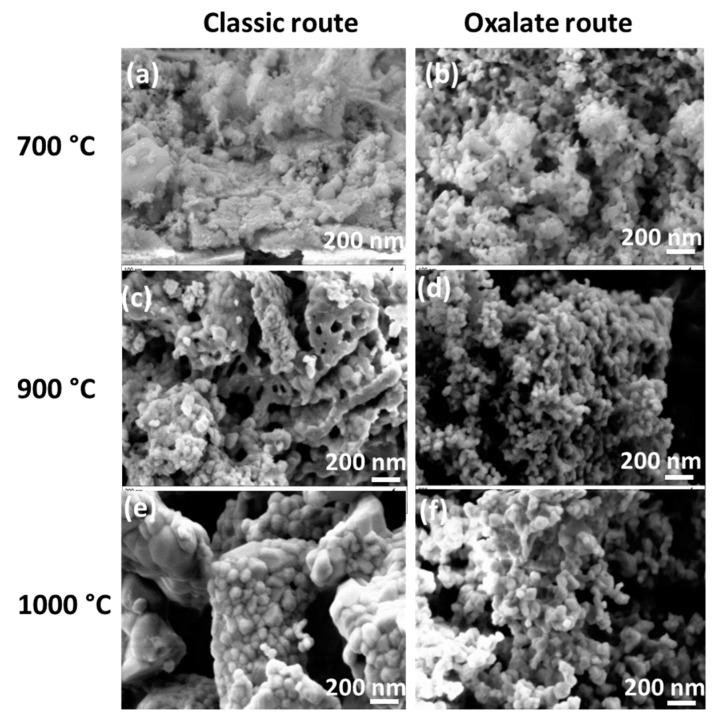
SEM images of Rh-doped samples BTR1-CL-700 (**a**), BTR1-OX-700 (**b**), BTR1-CL-900 (**c**), BTR1-OX-900 (**d**), BTR1-CL-1000 (**e**) and BTR1-OX-1000 (**f**).

**Table 1 nanomaterials-09-01697-t001:** Phase compositions of the different samples produced in this work.

Temperature (°C)	Classic Route		Oxalic Route	
	Undoped	Rh-Doped	Undoped	Rh-Doped
**700**	BaCO_3_	BaCO_3_		
	BaTiO_3_,	BaTiO_3_	BaTiO_3_	BaCO_3_
	BaTi_2_O_5_	BaTi_2_O_5_		BaTiO_3_
	TiO_2_ rutile	TiO_2_ rutile		
**900**	BaCO_3_	BaTiO_3_		
	Cubic BaTiO_3_	BaTi_2_O_5_	BaTiO_3_	BaTiO_3_
	BaTi_2_O_5_			BaCO_3_
	TiO_2_ rutile			
**1000**	BaTiO_3_	BaTiO_3_		
	BaTi_2_O_5_	BaTi_2_O_5_	BaTiO_3_	BaTiO_3_
		Ba_3_RhTi_2_O_9_	BaTi_2_O_5_	Ba_3_RhTi_2_O_9_
				BaCO_3_

**Table 2 nanomaterials-09-01697-t002:** Elemental compositions of undoped and doped samples prepared at 700 °C obtained from energy-dispersive X-ray spectroscopy (EDX).

	BT-OX-700			BT-CL-700			BTR1-OX-700			BTR1-CL-700		
	1	2	3	1	2	3	1	2	3	1	2	3
**Ti (%)**	21.99	20.92	21.13	15.12	20.34	4.08	14.03	14.13	15.73	22.31	16.08	18.92
**Ba (%)**	19.79	18.04	18.72	14.58	30.07	15.32	15.21	15.45	16.77	19.79	14.62	19.24
**Rh (%)**							0.45	0.45	0.49	0.44	0.40	0.47
**C (%)**				17.66	18.82	29.47	9.43	5.89	4.03	9.03	9.17	9.28
**O (%)**	58.22	18.04	60.15	52.64	30.77	51.13	60.88	64.08	62.98	48.17	59.73	52.09
**Total**	100	100	100	100	100	100	100	100	100	100	100	100

**Table 3 nanomaterials-09-01697-t003:** Specific surface area values of the as synthesized materials at 900 and 1000 °C.

Samples	Specific Surface S_BET_ (m^2^)
BT-CL-900	3 ± 0.007
BT-OX-900	12 ± 0.03
BTR1-CL-900	4 ± 0.009
BTR1-OX-900	17 ± 0.02
BT-CL-1000	1 ± 0.008
BT-OX-1000	5 ± 0.04
BTR1-CL-1000	1 ± 0.005
BTR1-OX-1000	2 ± 0.01

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
