# Peer review of "Impact of Oxalate Ligand in Co-Precipitation Route on Morphological Properties and Phase Constitution of Undoped and Rh-Doped BaTiO3 Nanoparticles"

_nanomaterials, 2019, doi:10.3390/nano9121697_

Round 1

Reviewer 1 Report

The authors elaborate on the influence of synthesis procedure of Rh-doped BaTiO3 on its phase purity and morphological properties. The basic problem with this manuscript is that, being an exploratory paper, it covers a very narrow range of preparation methods of the investigated compound. Only one new synthesis route, oxalic, is compared with the so called modified classical route. It can be easily found in the relatively recent literature that other wet chemistry ways of synthesis are also possible. Therefore, in my opinion, it is a not completed work and other potential synthesis routes should be explored to draw meaningful conclusions. 

Moreover, the selected reference synthesis procedure appears to be somewhat flawed. At no point a pure phase BaTiO3 was obtained, which raises the question of validity of this method.

Additional points include:

it is not explained in the abstract what a classical route means; rhodium not Rhodium; optical properties are not mentioned in the introduction; some easily found references to the other reports on the synthesis of this material:  https://doi.org/10.1021/am405293e, https://doi.org/10.1039/C5RA20044J; it is not explained how the elemental analysis with the EDX was performed quantitatively and what is the accuracy and precision of this evaluation; what is the definition of crystallinity - what the authors mean by that? what are the estimated errors of lattice parameters? since it is an important factor in the discussion it should be taken into account; "dissociation of Rh3+ -ions out of the BaTiO3 lattice" - I think the "dissociation" word is used incorrectly here; it would be beneficial for the reader to mark the peaks in the Raman spectra in Fig. 5, and the legend within Fig. 5 is not defined; Please elaborate on the meaning of "The presence of a ligand (e.g. in this case oxalate) avoids the stacking of the particles by steric hindrance during decomposition." How is that statement justified? Fig. 8 is unnecessary, and it is of poor quality; the accuracy of the specific surface area values is missing, especially for such a small values; "The doping mechanism of rhodium is debatable and can be assumed to be substitution of Ti4+ by Rh3+ for both preparation routes." - in my opinion it is not a mechanism but an effect. Still, why can it be assumed so?

Author Response

Thank you for your comments. Please find our replies and instructed corrections at the attached file.

Reviewer 2 Report

The article titled “Impact of synthesis route on microstructural and physical properties of undoped and Rh-doped BaTiO3 nanoparticles” by Roussin Lontio Fomekong et al. explains the synthesis of BaTiO3 nanoparticles (NP) using 2 different synthesis pathways, a classical one using a solid state reaction and a oxalate mediated route of co-precipitation. The final calcination step is carried out at 3 different temperature (700°C, 900°C, 1000°C) and the effect of a Rh-doping is investigated. The products of the in total 12 synthesis versions were characterized for their identity, composition, morphology and polymorphism.

The obtained data is appropriately discussed and compared to literature values of similar systems or analysis techniques with sufficient references given. The paper is well structured and takes the reader through the different aspects of the synthesis and structural parameter. The authors mention gas sensors as potential application for the synthesized materials, however, the draft is lacking in defining the ideal parameters for NP size, material morphology and polymorph type of the material to be used as a gas sensor. What role would the dopant play in a hypothetical application?

I would recommend the article for publication in the journal Nanomaterials after minor corrections.

Detailed comments (scientific points):

Materials and Methods: The authors should state the company with city and country for purchases and machines.

L121: “… through a 50x objective…” please state some objective specs like numerical aperture

L133: “… Figure 1a the undoped powders synthesized via oxalate route …” wrong reference, fig1a is the classical route.

Figure4: “Lattice parameters for samples prepared by oxalic route (a) and by classic route (b)” Are the lattice parameters given for the barium titanate phase and the respective Ba3RhTi3O9? Could the authors please clarify? It is not clear why a plot with bars is favourable over normal symbols.  Also, missing articles and conjunction “Lattice parameters for samples prepared by the oxalic route (a) and by the classic route (b), respectively.”

L196: “… spectra are more sensitive to short range ordering structure.” The authors might want to give a reference for that statement and the statement in L203 “room temperature P4mm phase.”.  

Figure 5: For better comparison the authors can bring the box surroundings of the graphs into contact. Thus, the labels and numbers of the X/Y axes would only be printed for the 2 most left (Y-axis) and the bottom row (X-axis). Also, marking some of the important bands named in the text could be highlighted. For this vertical line could span 2 graphs for better overview. It seems rather confusing plotting the long sample names. The authors might want to name the rows 700 °C / 900 °C 1000 °C for clarity. The remaining numbers in the graphs could be a little larger in size.

Figure6/7: The authors might consider labelling the rows (700 °C / 900 °C / 1000 °) and columns (classical / oxalate) for clarity. The white bottom bit of every pane in the 2 figures can be cut out, since those information can rather be given in the materials and methods section.

Figure 8: Mention in the caption that all given data is at 700°C. It is further very hard to see the elemental symbols and thus to understand the data in the graph.

Table 2: Is there any such data for the 900 and 1000°C preparations? Maybe put this at least in the supporting information. Would it be beneficial to average the 3 individual measurements to give average values of the sample? The values of sample BT-CL-700 number 2 and 3 are very different to the number 1 and the other graphs. Is there and explanation for this?`

L307: “…presented in Table 2.” Wrong reference. It should say table 3.

Detailed comments (spelling/wording)

L15: “…specific application like gas sensor, the…” missing plural “…specific application like gas sensors, the…”

L18: “different temperature. XRD, Raman” missing plural…“…different temperatures. XRD, Raman…”

L29: “... obtained by oxalic route, while classic route…” missing article “…obtained by the oxalic route, while the classic route…”

L31: “… The presence of oxalate ligand with its…” missing article “… The presence of the (an) oxalate ligand with its …”

L32: “… steric hindrance, that promotes the uniform…” no comma “… steric hindrance that promotes the uniform…”

L33: “… be responsible of the great difference…” wrong preposition “… be responsible for the great difference…”

L49: “… piezoelectric properties, BaTiO3 is a well-known…” missing subscript “… piezoelectric properties, BaTiO3 is a well-known…”. Please also check for subscripts in L295.

L57: “… high temperatures (>= 1000 °C). For instance…” consider use other symbol like “≥ 1000 °C”

L59: “… sol- gel, co-precipitation and etc. techniques…” awkward wording “… sol- gel, co-precipitation and other techniques…”

L71: “… modified solid-state route (thereafter named as ‘classic route’). …” awkward wording “… modified solid-state route (hereafter named as ‘classic route’). …”

L94: “Same processing steps are employed for synthesis of…” missing article “The same processing steps are employed for synthesis of…”

L102: “… previously prepared (metal) cationic …” the word in parentheses is unnecessary, consider cancelling “… previously prepared cationic …”

L104: “… was filtered and dried in oven at 80 °C yielding …” article missing “… was filtered and dried in an /the oven at 80 °C yielding …”

L105: “… combustion boat holder at 700°C, 900 °C and …”… missing space “combustion boat holder at 700 °C, 900 °C and …” please also check the missing space before the °C sing in L134, 137, 157, 159.

L111/112: “… which has a Bragg-Brentano geometry and equipped with Cu-Kα radiation …” awkward wording “… which has a Bragg-Brentano geometry using Cu-Kα radiation …”

L112/113: ”… For the experiment, the as prepared material was pressed …” missing plural “”… For the experiments, the as prepared materials were pressed …”

L114: “… scan step of 0.02 (2Ɵ) and a 2 s step−1. …” missing superscript “… scan step of 0.02 (2Ɵ) and a 2 s step−1. …”

L116: “… diffractograms with the program EVA from BRUKER AXS …” preposition “… diffractograms with the program EVA by BRUKER AXS …”

L120: “… spectrometer from Bruker under 532 nm and 0.2mW power laser …” awkward wording “… spectrometer by Bruker using 532 nm and 0.2mW power laser …”

L125: “… particulate Platine layer to avoid charge effect. …” missing plural “… particulate Platine layer to avoid charge effects. …”

L139: “… That means that oxalic route … ” wording can be improved and missing article “… This means that the oxalic route … ”

L144: “… the intensity of peak increases…” awkward wording “… the XRD peak intensity increases…”

L156: “No evidence of Rh phase was encountered …” missing article “No evidence of an Rh phase was encountered …”

L163/164: “In other words, Bragg diffraction peak of powders calcined at this temperature is nearly coincided with the undoped…” missing article and plural as well as wrong wording “In other words, the Bragg diffraction peaks of powders calcined at this temperature are nearly coincident with the undoped…”

L172/173: “… composition of doped sample are similar to that obtained by oxalic route, at 700 °C and 900 °C classic route leaded again to less pure powder …” awkward wording “… composition of doped sample is similar to that obtained by oxalic route, at 700 °C and 900 °C following the classic route leads again to less pure powders …”

L175: “… and there is no decrease of peak intensity like in oxalic …” missing articles “… and there is no decrease of the peak intensity like in the oxalic …”

L177: “… increase of lattice parameter is observed…” missing article “… increase of the lattice parameters is observed…”

L181: “That is not the case with classic route…” missing article “That is not the case with the classic route…”

L205: “However the samples prepared by classic route …” missing comma and article “However, the samples prepared by the classic route …”

L221: “… that is significantly different to that of the …” wrong preposition “… that is significantly different from that of the …”

L231: “At 900 °C some peaks …” missing comma “At 900 °C, some peaks …”

L234: “… the peaks at 725 cm-1 are already intense…” wrong plural “… the peak at 725 cm-1 is already intense…”

L242: “… leads well-defined and homogeneously…” missing preposition “… leads to well-defined and homogeneously…”

L271: “… (EDX) 271 is analytical technique …” missing article “… (EDX) 271 is an analytical technique …”

L275: “… EDX of 4 samples (BT-OX-700, BT-CL-700, BTR1-OX-700 and BTR1-OX-700) took …” wrong conjugation “… EDX of 4 samples (BT-OX-700, BT-CL-700, BTR1-OX-700 and BTR1-OX-700) taken …”

L290: “… For undoped sample prepared by oxalic route, …” missing plural and article “… For the undoped samples prepared by the oxalic route, …”

L331: “… barium titanate such gas sensor which will …” missing word and plural “… barium titanate such as gas sensors which will …”

L335: “… different temperatures (700 °C, 900 °C and …” wrong wording “… different temperatures (700 °C, 900 °C or …”

L345: “… morphology and nanoparticles sizes are observed with oxalic route …” wrong plural and missing article “… morphology and nanoparticle sizes are observed with the oxalic route …”

L351: “… application such as gas sensor.” Missing plural “… application such as gas sensors.”

Author Response

Thank you for your comments. Please find our replies and the instructed corrections in the attached file.

Round 2

Reviewer 1 Report

The authors have improved the manuscript and clarified the aim of the study. With these modifications I recommend the paper for publication after additional changes:

authors write that the oxalate anion act as a buffer (puffer?) and prevent stacking of the particles during thermal treatment. Firstly, they use the term ligand, which may be correctly used for the precursor solution, where there exists a central atom (ion). (The term ligand shouldn't be used for molecules adsorbed on the surface of nanoparticles.) However, during the thermal treatment the oxalate ions decompose, and probably at much lower temperatures than the final calcination temperature (maybe add TGA). At the highest temperatures employed in the synthesis, the sintering will occur. At that time there are no oxalates present. Could the authors elaborate more on the role of the oxalates on the final product formation route? Since it is the main subject of the study, it is quite important.  authors added errors to the SSAs, but it's quite hard to believe that they are so small! In addition significant figures in the SSA values should be adjusted. in conclusions, the authors state that a well-defined spherical morphology is obtained. However, there are no analyses to support this statement. In my opinion, the provided SEM figures does not allow to draw this conclusion.

Author Response

Please find the reply to your comments in the attached file.

We have also generated data files from BET measurements of the Oxalate and classic route produced powders and tried to attach them hereby. There might be some Problems with the upload. If needed we can submit them later again.
